# Unexpected Differences in the Speed of Non-Malignant versus Malignant Cell Migration Reveal Differential Basal Intracellular ATP Levels

**DOI:** 10.3390/cancers15235519

**Published:** 2023-11-22

**Authors:** Bareun Kim, Anthony T. Lopez, Indhujah Thevarajan, Maria F. Osuna, Monica Mallavarapu, Boning Gao, Jihan K. Osborne

**Affiliations:** 1Department of Pharmacology, University of Texas Southwestern Medical Center, 5323 Harry Hines Blvd., Dallas, TX 75390, USA; bareun.kim@utsouthwestern.edu (B.K.); anthony.lopez@utsouthwestern.edu (A.T.L.); indhujah.thevarajan@utsouthwestern.edu (I.T.); maria.osuna@utsouthwestern.edu (M.F.O.); monica.mallavarapu@gmail.com (M.M.); 2Hamon Center for Therapeutic Oncology, University of Texas Southwestern Medical Center, 5323 Harry Hines Blvd., Dallas, TX 75390, USA; boning.gao@utsouthwestern.edu

**Keywords:** cell migration, cancer cell migration, normal epithelial cell motility

## Abstract

**Simple Summary:**

Cell motility involves the coordination of several complex activities such as cytoskeletal rearrangements and extracellular matrix remodeling that require considerable amounts of energy in the form of adenosine triphosphate (ATP). Disease states like tumorigenesis co-opt many of the cellular processes described above to facilitate metastasis, the primary cause of cancer-related death. Many of the genes required for the locomotion of cancer cells are similarly important for normal adult stem cell regeneration and wound healing. Since proper wound healing and regulated regeneration are critical to maintaining homeostasis into adulthood, understanding the differences between normal and cancer cell motility is essential in developing targeted therapies. We screened human normal and cancer cell lines derived from the lung and breast for migratory potential. To our surprise, normal human lung epithelial cells migrated faster than the lung cancer cells, representing a paradigm shift in the development of future targeted metastatic therapies.

**Abstract:**

Cellular locomotion is required for survival, fertility, proper embryonic development, regeneration, and wound healing. Cell migration is a major component of metastasis, which accounts for two-thirds of all solid tumor deaths. While many studies have demonstrated increased energy requirements, metabolic rates, and migration of cancer cells compared with normal cells, few have systematically compared normal and cancer cell migration as well as energy requirements side by side. Thus, we investigated how non-malignant and malignant cells migrate, utilizing several cell lines from the breast and lung. Initial screening was performed in an unbiased high-throughput manner for the ability to migrate/invade on collagen and/or Matrigel. We unexpectedly observed that all the non-malignant lung cells moved significantly faster than cells derived from lung tumors regardless of the growth media used. Given the paradigm-shifting nature of our discovery, we pursued the mechanisms that could be responsible. Neither mass, cell doubling, nor volume accounted for the individual speed and track length of the normal cells. Non-malignant cells had higher levels of intracellular ATP at premigratory-wound induction stages. Meanwhile, cancer cells also increased intracellular ATP at premigratory-wound induction, but not to the levels of the normal cells, indicating the possibility for further therapeutic investigation.

## 1. Introduction

The complex process of cell migration usually initiates with cytoskeletal rearrangements and signaling cascades that trigger gene expression and shifting metabolic requirements. Numerous disease states, including the metastatic spread of aggressive tumors, alter cell migration. Several studies have described various modes of movement within malignant and non-malignant cells as both individual cells and as collective sheets.

Despite the disease state or mode of cell migration, the energy demands of cytoskeletal rearrangements, volume regulation, anoikis resistance, and extracellular matrix remodeling are elevated in migrating cells compared with confluent/compressed cells [1,2,3,4,5,6,7]. The normal mammary epithelial cell line, MCF10A, and the metastatic breast cancer cell line, MDA-MB-231, have been used concurrently and alone as exemplars of in vitro cell migration in hundreds of studies. Utilization of the MCF10A and MDA-MB-231 cell lines have led to a deeper understanding of the five **W**’s of cancer metastasis including *what* the extracellular matrix is [5]; *when* growth factor (e.g., TGFβ) signaling is unfavorable for tumorigenesis yet pro-metastatic [8,9]; *how* metastatic cells remodel distant organs [10,11]; and, finally, the debatable conclusion of *why* faster cancer cell migration is an indicator of metastatic potential [12,13,14,15].

Robust cell migration (quantified by speed, persistence, and track length) has been described as a hallmark of cancer metastasis [13,16]. Initially, we screened 15 cell lines: 12 lung lines and three mammary/breast lines derived from both non-malignant and malignant cells via scratch wound assay, to identify migrators for further experimentation in an unbiased manner—fast and slow. Our data revealed an interesting result; non-malignant cells from the lung regardless of the media used for migration closed the wound faster than the lung cancer cells. This led us to modify our inquiry to ask “Why are normal cells faster at wound closure than cancer cells if migration speed is a hallmark of cancer metastasis?”.

Cell migration/invasion necessitates a considerable expenditure of energy in the form of ATP [6,17]. We discovered that non-malignant cells (of both the lung and mammary) had greater basal ATP levels than the cancer cells at wound induction. Additionally, all cell lines differentially increased ATP content at the onset of wound induction, indicating a critical demand for energy at wound induction during premigratory states regardless of disease state.

## 2. Materials and Methods

### 2.1. Cell Lines

Seven normal human bronchial epithelial (HBEC) and six lung carcinoma cell lines were screened for their ability to migrate on collagen and to invade Matrigel. All cancer cells were grown in RPMI 1640 supplemented with 5% FBS, maintained at 37 °C in an atmosphere of 5% CO_2_. All normal, non-tumorigenic cells were grown in keratinocyte serum-free media kit (KSFM; Gibco, Waltham, MA USA kit catalog no. 17005042), except for MCF10A (grown in MEGM, Lonza, Cambridge MA USA kit Catalog no. CC-4136), and maintained at 37 °C in an atmosphere of 5% CO_2_. The HCC4017 cell line was additionally adapted to non-malignant cell media, KSFM, with and without the addition of 2% FBS. Lung cancer cells and human bronchial epithelial cells (HBEC), unimmortalized and immortalized, were derived, deposited, and maintained by Hamon Cancer Center, UTSW Dallas TX. Human cell lines were authenticated to confirm origin, and all cell lines were pre-treated with Plasmocin until confirmed to be free of mycoplasma (Myco-Sniff-Rapid^TM^ Mycoplasma Luciferase Detection Kits, MP Biomedicals, Santa Ana, CA, USA) before use.

### 2.2. Materials

RPMI 1640 and Trypsin-EDTA were purchased from Sigma-Aldrich (St. Louis, MO, USA). Keratinocyte SFM (1X), PBS pH7.4 (1X), and Gentamicin Reagent Solution were purchased from Thermo Fisher Scientific (Grand Island, NY, USA). Cell culture plates were purchased from Corning Life Sciences (Corning, NY, USA) and Thermo Fisher Scientific (Grand Island, NY, USA). Plasmocin treatment reagent was purchased from InvivoGen (San Diego, CA, USA). A 96-well Wound Maker and IncuCyte CytoATP Lentivirus kit were purchased from Sartorius (Göttingen, Germany). ADP/ATP Ratio Assay Kit was from Abcam (Boston, MA, USA). Myco-Sniff-RapidTM Mycoplasma Luciferase Detection Kits was purchased from MP Biomedicals (Santa Ana, CA, USA). 

### 2.3. In Vitro Scratch Assay Quantification

All cell lines tested (3.5–4 × 10^4^ cells/well) were seeded onto a collagen-1- or Matrigel-coated 96-well ImageLock tissue culture plate (Sartorius) and incubated at 37 °C with 5% CO_2_ incubator for 24 h. The wounds were made by the 96-well Wound Maker (Sartorius Göttingen, Germany). The wounded cells were washed 3 times with 1XPBS plus gentamycin (Gibco Waltham, MA USA) to remove the detached cells before imaging. Cancer and normal cells adapted to non-growth condition media were adapted for 48–72 h prior to wounding and at wound induction.

### 2.4. Physical Measurements of HBEC30UI, HBEC30KT, and HCC4017 on LIVECYTE

Quantification of individual cell: dry mass, doubling time, radius, sphericity, volume, track length, and speed were all performed on the Livecyte kinetic cytometer system. With the use of their quantitative phase imaging (QPI) technique known as Ptychography, cells were visualized with ultra-low-powered laser to track proliferation, shape, dimensions, and migration. Cells from each cell line were plated in duplicate at a density of 40,000/well. Raw data were analyzed using GraphPad PRISM.

### 2.5. ATP/ADP Ratio Assay Kit

ATP/ADP ratios of unscratched and wound-induced cells were measured by ADP/ATP Ratio Assay Kit (#ab65313, abcam). Cell lysates were mixed with nucleotide-releasing buffer and incubated for 5 min at room temperature with gentle shaking. Then, 100 μL prepared reaction mix was added and the background luminescence was read (Data A); 50 μL sample was added; and, after 2 min, the luminescence was read (Data B). To measure ADP levels in the cells, the samples were read again (Data C); then, 10 µL of 1× ADP converting enzyme was added and the samples were read again after approximately 2 min (Data D). ATP/ADP ratio equation: ATP measurement/(ADP initial—ADP converted to ATP). Ratios were calculated from the ATP/ADP ratio of no scratch subtracted from ATP/ADP ratio of T = 0 scratch.

### 2.6. Measurement of Intracellular ATP Dynamics

ATP levels are measured in live cells using an ATP-binding FRET indicator provided by the IncuCyte CytoATP Orange Lentivirus Reagent kit (catalog #4772; purchased from Sartorius). Analysis was performed using the SX5 Metabolism Optical module and the ATP analysis software. Upon binding of cytoplasmic ATP conformational changes occurring in the ATP-binding sensor utilizing dual excitation (535X, mKOk orange; 485X, cpmEGFP green) and fixed emission of 578M, increasing amounts of ATP lead to increased transmission and emission at 575M, which is used to evaluate dynamic changes when compared with a non-ATP binding sensor. Cancer, normal/immortalized, and normal/primary cells were infected with the CytoATP binding or non-binding lentivirus followed by puromycin selection. After wound initiation, cells were visualized migrating into wound with the IncuCyte (model SX5).

### 2.7. Statistical Analysis

All graphs and statistical analyses were conducted using GraphPad Prism 8 or 10. Statistical significance was assigned to data with the following criteria: * *p* < 0.05 and N representing individual experiments greater than three, except for tests of physical parameters performed on isogenic cells. S.D., standard deviation; S.E.M., standard error of the mean. Data were analyzed using unpaired *t*-test, one-way ANOVA, and two-way ANOVA; ** p* < 0.05, ** *p* < 0.01, *** *p* < 0.005, **** *p* < 0.0001.

## 3. Results

Non-malignant cells migrate faster than cancer cells;Irrespective of media, normal cells outcompete cancer cells;Physical differences do not account for faster migration speed;Increased ATP demands at wound induction;ATP levels oscillate throughout wound resolution in normal cells.

### 3.1. Non-Malignant Cells Migrate Faster Than Cancer Cells

Recent evidence has indicated that approved FDA drugs, including cisplatin and paclitaxel, can promote metastasis and drug resistance in preclinical models [18,19,20]. Given these new revelations, we sought to develop a screen for migratory potential (Figure 1 and Figure 2). First, we screened malignant cell lines isolated from primary and metastatic tumors of lung and breast, together with their non-cancerous counterparts (as controls) for the ability to migrate and/or invade collagen/Matrigel (Figure 1 and Figure 2). Measuring the time for wound density to reach a defined percentage (Time for Density to reach 80% and 40%—TD80 and TD40), gave us a quantifiable readout of migration potential (Figure 1 and Figure 2b). What we detected unexpectedly was that the non-malignant cells of the lung had significantly higher migratory potential than tumor cells from their respective tissues (Figure 1b,c). We screened non-malignant, primary and immortalized, lung bronchial epithelial cell lines previously established from biopsies of patients with and without lung cancer [21,22,23]. Primary cells were cultured and transfected with cyclin-dependent kinase 4 (CDK4) and human telomerase reverse transcriptase (hTERT), resulting in cultures that did not undergo senescence. Immortalized cells did not undergo malignant transformation in vitro due to their inability to form colonies in soft agar, nor did they form tumors in vivo in immune-compromised mice [21]. Experimentally, we chose to focus on the isogeneic lung cell lines isolated from the same person. The HCC4017 cells were derived from the tumor-burdened lung, while the primary HBEC30UI (30UI) and immortalized HBEC30KT (30KT) were from the normal contralateral lung of the same patient [23]. These isogeneic cells were chosen not only for their genetic similarity but also for their close doubling times. The timing of migration experiments was designed to exclude gross complications due to proliferation (Figure 1d). Interestingly, we found that both non-malignant cell lines, 30KT and 30UI, migrated significantly faster than the cancer cell line HCC4017 of the same person (Figure 1e,f). Predictably, we observed that the non-cancerous MCAF10A cell line was significantly slower than the breast tumor cell line, MDA-MB-231 (Figure 2b,c), as demonstrated in previous studies [13,14]. Based on these observations, we redirected our investigation of “the differences between how non-malignant and malignant cells migrate” to “why are normal cells faster at wound closure than cancer cells”.

### 3.2. Irrespective of Media, Normal Cells Outcompete Cancer Cells

RPMI 1640 growth media (supplemented with 5% fetal bovine serum, FBS) were used for maintenance of all our lung cancer cell lines, while all our normal bronchial epithelial cells were maintained in keratinocyte serum-free media (KSFM). Although our initial screen for migration potential was conducted in cancer cells grown in RPMI 1640 supplemented with 5% FBS and normal cells grown in KSFM, we acknowledge that different media have vastly different components. Thus, in order to normalize extrinsic factors contribution to the varying cell speeds, we adapted our lung cancer cells to KSFM with varying concentrations of FBS. While KSFM with no FBS-supported HCC4017 migration cells did not proliferate, HCC4017 cells grown in KSFM plus 2% FBS supported proliferation and migration (Figure 3a). Interestingly, we found that normal cells adapted for migration experiments in KSFM plus 2% FBS, or RPMI 1640 with 5% FBS, were still able to close wounds (albeit, not statistically significant in RPMI 1640 with 5% FBS) faster (Figure 3b). Closer examination of wound closure dynamics indicated variations of both normal and cancer cells in RPMI 1640 with 5% FBS (Figure 3b,c).

### 3.3. Physical Parameters Do Not Account for Faster Migration Speed

To address our new query, we first investigated whether physical differences in mass, volume, or shape could account for the increased migration potential of the non-malignant cells (Figure 4). Between 1000 and 3200 individual cells were tracked, graphing only those cells that remained from the first to the last frame (A1 Appendix A). We found that the speed of the individual non-malignant cells, though not statistically different from each other, were significantly faster than that of the malignant cells (Figure 4f). No gross differences in mass, shape (sphericity), or dimensions (radius, volume) were found between the normal (either 30KT or 30UI) and cancer cells (Figure 4b–e). Statistically significant differences in mass, sphericity, and volume observed between 30KT and 30UI were attributed to the immortalization process (addition of constitutive expression hTERT and CDK4), as the quantification of the same dimensions between 30KT and HCC4017 were not statistically significant. Taken together, these data suggest that the differences observed in the speed of malignant cells and non-malignant cells were not due to their physical parameters.

### 3.4. Increased Intracellar ATP Demands at Wound Induction

A considerable expenditure of energy in the form of ATP is needed for cell migration. Differences between how normal and cancer cells meet this high demand for ATP during migration are currently not well understood. Intracellular ratios of ATP-to-ADP have been traditionally used as a readout of cellular energy, where energy is derived from the hydrolysis of ATP to ADP [24,25]. We assessed energy changes during migration, beginning with premigratory-wound induction via two different ATP assays (Figure 5a). We measured the changes in ATP/ADP ratios in lysates from confluent/no scratched (NS) and premigratory-wound-induced cells (T = 0); there was an increase in ATP/ADP ratios from NS to T = 0 (Figure 5b), indicating an acute energy demand. Next, we constructed cell lines stably expressing a lentiviral FRET-based ATP sensor in several non-malignant and malignant cells (Figure 5c). Like the first assay, most of the cells showed increased intracellular ATP levels going from NS to T = 0. Subsequently, in the non-malignant cells of the breast and lung, there was also a higher-fold change in the ATP/ADP ratios than in their respective cancer tissues (Figure 5d,e). Use of cells stably expressing the CytoATP sensor allowed for tracing intracellular ATP (iATP) levels throughout migration (Figure 5d,e). The isogeneic pair 30KT and HCC4017 revealed a spike in the iATP levels that occurred at the premigratory-wound induction stage (T = 0), which resolves and stabilizes over the course of wound closure (Figure 5e). Likewise, the other cell lines show the same results: there is an acute energy demand that accompanies wound induction in both normal and cancer cells (Figure 5d,e). 

### 3.5. ATP Levels Oscillate throughout Wound Resolution in Normal Cells

Various modes and morphologies of cancer and normal epithelial cell migration have been described ranging from single cell migration (e.g., amoeboid or mesenchymal state) to collective migration (e.g., mesenchymal chains, clusters, and multicellular sheets undergoing biophysical unjamming) [26,27,28,29,30]. Recently, hybrid states of collective migration have also been reported describing the ability of invading cells to adopt leader–follower hierarchies [26,27,28,31,32]. Mechanisms responsible for the plasticity of migratory modes range from variations in genetic/epigenetic signals; the cell state of premalignant lesions; and, more recently, cellular energetics [33,34,35]. Previous literature suggests a need to maintain certain thresholds of ATP to maintain leader positions during collective migration [35]. While our results indicate an initial burst of iATP at wound induction (Figure 5b,c), we also found that iATP levels were stabilized throughout migration until wound resolution, where there was an increase in iATP in the normal cells upon wound closure that correlated with the speed of the cells (Figure 6a).

Variations in iATP levels in singular cells were detected in the immortalized (30KT) cells compared with the primary unimmortalized cells (30UI) in confluence (Figure 6b) that lessened throughout migration (Figure 6c). Additional dissimilarities in migration modes were also observed, whereby the 30UI cells moved primarily as collective sheets led by discrete “leader” cells while the 30KT cells exhibited single and pockets of collective migration like the HCC4017 cancer cells. We attributed the migration mode differences in the normal cells to the immortalization process. In our model, we observed no gross increase in iATP dynamics specifically at the migratory edge of the wound (Figure 6c).

## 4. Discussion

Previous reports have highlighted the vast discrepancies between plasticity, proliferation rates, and the metabolic needs of malignant cells compared with non-malignant cells. Few studies, however, have systematically compared these same parameters side-by-side during migration, resulting in a lack of understanding of what the basic requirements are for cancer and normal cell locomotion that may be exploited therapeutically [5,15,30,36,37,38,39]. We conducted wound-healing/migration screens of normal and cancer cells isolated from breast and lung and found that normal lung cells closed wounds faster than lung cancer cells. Our working hypothesis for potential differences led us to investigate intracellular ATP. The function of ATP as a switch for increased cell migration has been well documented primarily in the brain and immune system [40,41,42,43,44,45,46]. These pleiotropic effects of ATP range from paracrine—indicator of injury, stimulator of immune response and repair, to autocrine—sensor and regulator of intracellular metabolic demand. Intracellular and extracellular ATP sensing carried out via the purinergic receptor family, which in addition to their modulation of calcium release act in an autocrine manner to subsequently produce more ATP increases and migration via cytoskeletal and matrix remodeling.

Here, we report that normal lung epithelial cells have higher intrinsic intracellular basal and wound-induced intracellular ATP levels than lung tumor cells. There have been several studies examining the consequences of ATP release and generation following matrix detachment, wounding/injury, or metastasis [5,6,17,37] of normal or cancer cells. Our study, however, focuses on the high-throughput comparison of pre- and post-wound induction in normal cells side-by-side with cancer cell counterparts with the hope of identifying intrinsic differences in migratory potential. These findings have broad-reaching implications for aggressive carcinomas from other non-breast tissues such as the colon and pancreas. Our data also rouse further important questions concerning ATP generation and its use in response to varied fuel sources or other metabolic needs during normal wound healing and throughout the metastatic cascade. Recent studies have both described heightened and suppressed Warburg effects during cancer cell migration; thus, additional investigations are critically needed to elucidate differential mechanisms in normal cell migration as well [6,17,35,47]. The therapeutic connections to various non-malignant lung diseases are also intriguing and range from tissue regeneration post-respiratory infection, autoimmune infiltration, and wound repair linked to asthma, pulmonary fibrosis, and acute or COPD (Chronic Obstructive Pulmonary Disease) to name a few.

## 5. Conclusions

Unbiased screening for the migratory potential of cells from both normal and cancer tissues of the lung revealed that normal bronchial epithelial cells (both immortalized and unimmortalized/primary) migrated faster than (1) established lung cancer cell lines and (2) the isogeneic tumor tissue established from the same patient. We were able to repeat data from countless other studies demonstrating the augmented migratory potential of metastatic breast cancer cells of the MDA-MB-231 compared with the normal immortalized MCF10A cell line, thus validating our experimental set-up. Further examination indicated higher basal levels of intracellular ATP in normal cells of both breast and lung, which increased upon wound-induction.

## Figures and Tables

**Figure 1 cancers-15-05519-f001:**
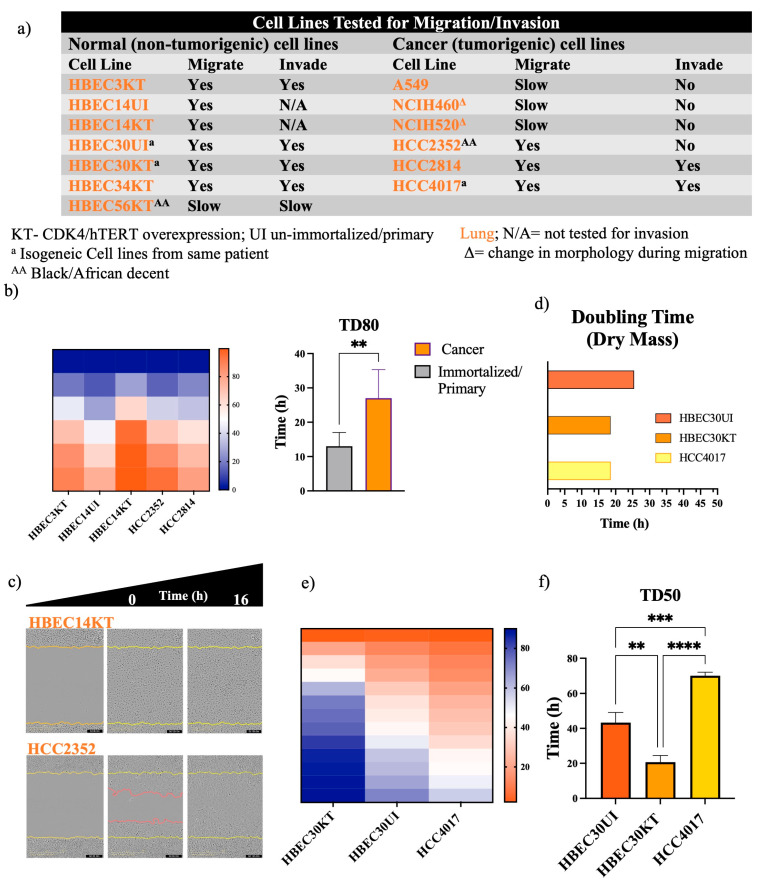
Non-malignant lung cells close wounds faster than cancer cells. (**a**) Table of lung cell lines tested for the ability to migrate on collagen and invade Matrigel. KT-CDK4/hTERT overexpression; UI, unimmortalized/primary ^a^ isogeneic cell lines from same patient; ^AA^ Black/African descent. Normal and cancer cells were plated in duplicate in a 96-well plate at a density between 35,000 and 50,000/well. Heatmaps for panel (**b**) represent the quantification of time to closure. Graph of panel (**b**) quantitates the time for wounds to reach a density of 80% (TD80) in lung. Wounds were initiated using the 96-well Incucyte^®^ Wound Maker. (**c**) Images were collected every 1–2 h on the SX5 Incucyte ^®^Live cell imaging instrument and analyzed using the Sartorius 96-well cell migration software application. Yellow lines in images indicate the initial wounded area; red lines represent cells moving into the wound. (**d**) Doubling times of the isogeneic cell lines chosen for further experimentation. Heatmaps for panel (**e**) represent the quantification of time to closure. (**f**) Quantitation time for wounds to reach a density of 50% (TD50) in isogeneic cell lines. Data were analyzed using unpaired *t*-test; ** *p* < 0.01, *** *p* < 0.0005, **** *p* <0.0001 Error bars represent +/-S.D. N is the quantitation of individual experiments. N = 3 for normal cells: HBEC3KT, 14UI, 14KT, 30UI, and 30KT; for cancer cells, N = 5 for HCC2352, N = 4 for HCC2814, and N = 3 for HCC4017.

**Figure 2 cancers-15-05519-f002:**
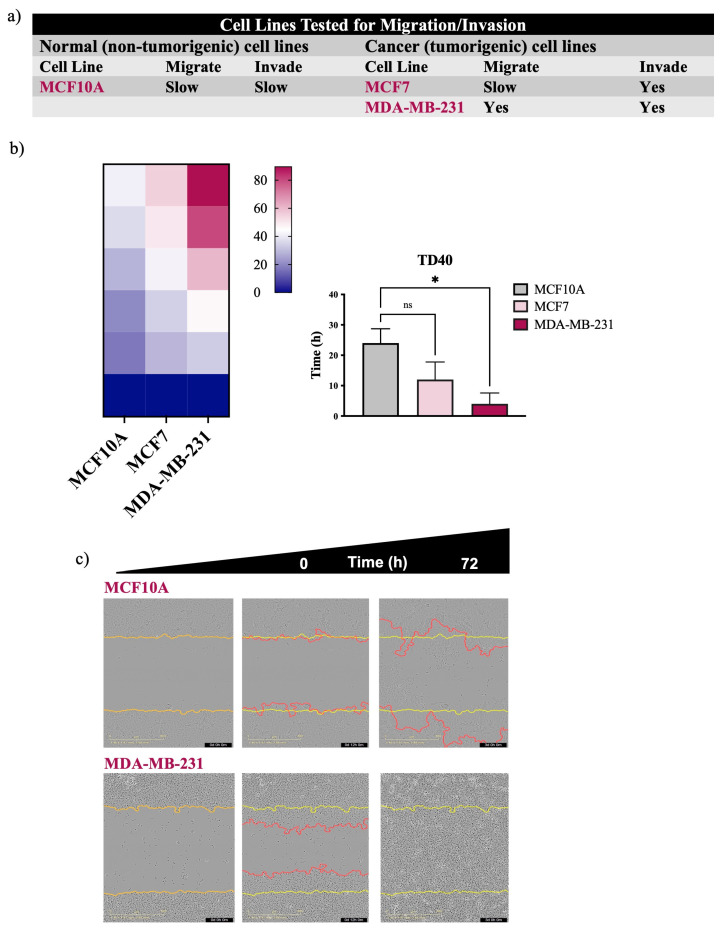
MDA-MB-231 achieves faster wound closure than MCF7 and MCF10A. (**a**) Table of breast/mammary cell lines tested for their ability to migrate on collagen and invade Matrigel. Cells were plated in duplicate in a 96-well plate at a density between 35,000 and 50,000/well. Heatmaps for panel (**b**) represent the quantification of time to closure. Graph of panel (**b**) quantitates the time for wounds to reach a density of 40% (TD40). (**c**) Images were collected every 1–2 h on the SX5 Incucyte^®^ Live cell imaging instrument and analyzed using the Sartorius 96-well cell migration software application. Yellow lines in images indicate the initial wounded area; red lines represent cells moving into the wound. Error bars represent S.E.M. N is quantitation of individual experiments; N = 3. Data were analyzed using one-way ANOVA; * *p* < 0.05 for MCF10A compared with MDA-MB-231, ns = not statistically significant.

**Figure 3 cancers-15-05519-f003:**
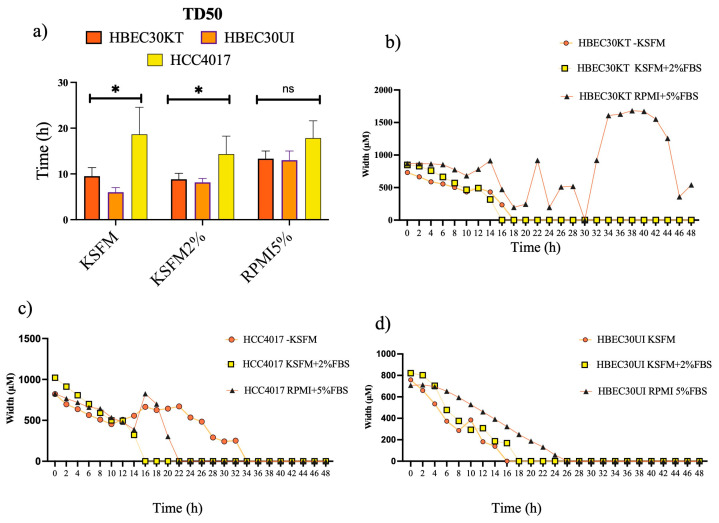
Irrespective of media, normal cells outcompete cancer cells. (**a**) Graph of isogeneic lung cells quantitating the time for wounds to reach a density of 50% (TD50) in various media. (**b**–**d**) Measurements of wound width (in μm) of each cell line in different media. Error bars in (**a**) represent individual experiments of N = 3 for each cell line +/-S.E.M. * *p* < 0.05 achieved using two-way ANOVA. n.s.= not statistically significant Wound measurements are representative of one experiment from (**a**).

**Figure 4 cancers-15-05519-f004:**
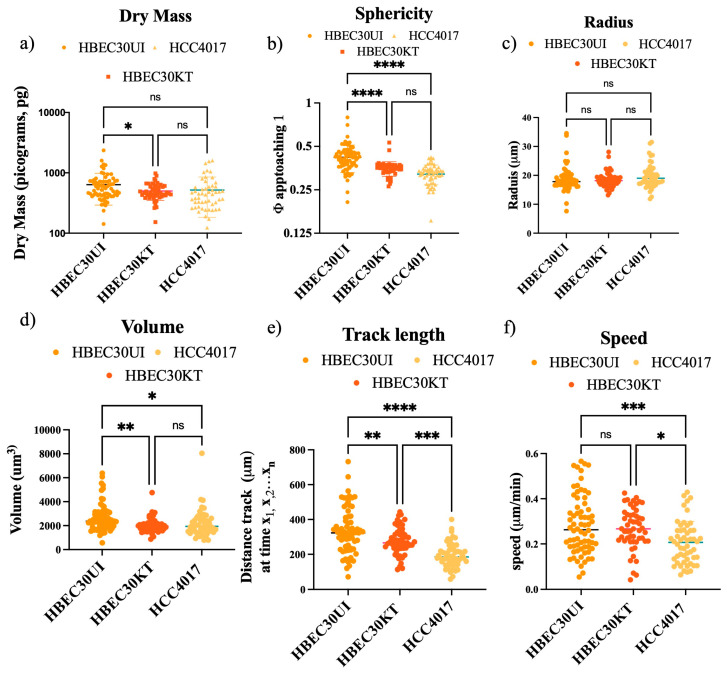
Physical parameters do not account for faster migration speed. Each cell line was plated at a density of 40,000/well. Wounds were initiated using the 96-well Incucyte^®^ Wound Maker. Images were collected every 30 min on the LiveCyte imager. A total of 45 frames were collected. A combination of 1000 (HBEC30UI), 2500 (HCC4017), and 3200 (HBEC30KT) cells were initially tracked and analyzed. The first 50 individual cells that were visible from frame 1 to frame 45 were graphed. Graphs represent 2 individual experiments run simultaneously with 3 replicates per cell line. Statistics of (**a**) dry mass, (**b**) sphericity, (**c**) radius, (**d**) volume, (**e**) track length, (**f**) speed were all achieved using S.D. one-way ANOVA; * *p* < 0.05, ** *p* < 0.01, *** *p* < 0.005, **** *p* < 0.0001, ns = not statistically significant.

**Figure 5 cancers-15-05519-f005:**
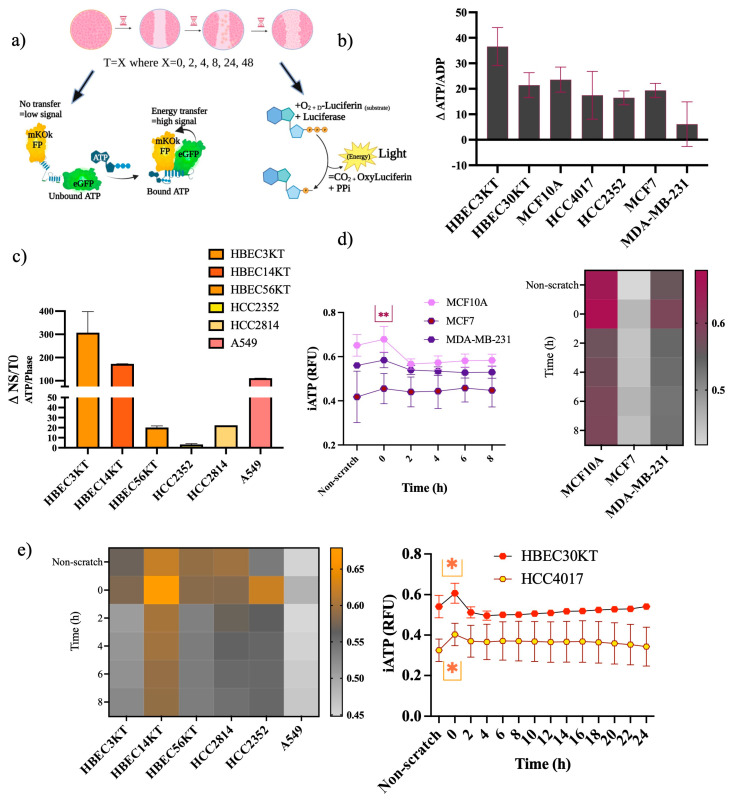
Increased ATP demands at wound induction. (**a**) Experimental scheme for ATP assessment. Experimental schematic created using BioRender (Toronto, ON, Canada). For (**b**–**e**), wounds were initiated using the 96-well Incucyte^®^ Wound Maker. Images were collected every 2 h on the SX5 Incucyte ^®^Live cell imaging instrument and analyzed using the Sartorius (96-well) SX5 Metabolism Optical module and the ATP analysis software. (**b**) Cells were plated to confluency, lysates extracted from non-scratch control cells (NS) and at wound induction (T = 0). ATP/ADP ratios of wound induction were subtracted from non-scratch controls (ΔATP/ADP). Error bars are representative of 2 out of 4 experiments. (**c**,**d**) Intracellular ATP measurements of cell lines. (**c**) Non-malignant and malignant cancer cells analyzed only for NS and T = 0. Error bars on (**c**,**d**) are of 3 independent experiments. Intracellular ATP measurements of (**e**) normal and cancer cells of the lung. Cells for (**c**–**e**) were all plated at a density of 40,000/well. Error bar graphs in (**e**) are N = 4 for 30KT and N = 3 for HCC4017; heatmaps represent standard deviation of N = 3 independent experiments; ** *p* < 0.01, ns = not statistically significant. Statistical significance was assessed using a two-way ANOVA for (**d**,**e**). Upper* indicates statistical difference between HBEC30KT and HCC4017 at wound induction and lower* indicates a statistical difference before wound induction.

**Figure 6 cancers-15-05519-f006:**
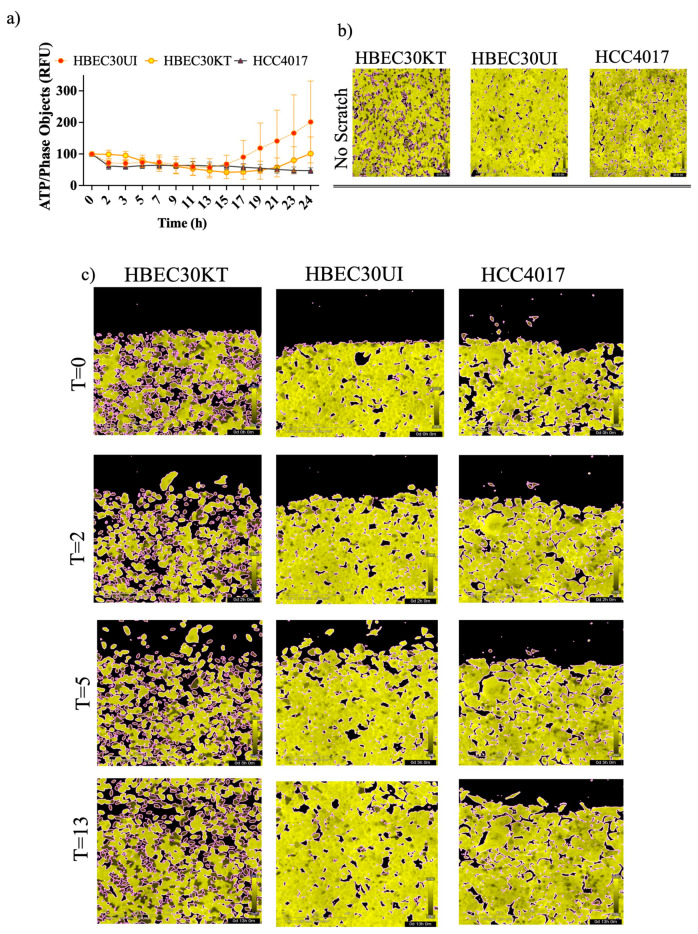
Increased ATP demands at wound induction. Isogeneic cells stably expressing the iATP construct were seeded at a density of 35–40,000/well. (**a**) Images of the FRET channel (fixed emission of 578M, Yellow) were used to quantitate iATP per individual cells migrating into the wound via phase then normalized to time zero. (**b**) Images taken in the FRET channel before wound (**c**) Images taken in the FRET channel at time 0, 2, 5, and 13 h post wound-induction.

## Data Availability

Data is contained within the article or Appendix A.

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
