# Peer review of "Unexpected Differences in the Speed of Non-Malignant versus Malignant Cell Migration Reveal Differential Basal Intracellular ATP Levels"

_cancers, 2023, doi:10.3390/cancers15235519_

Round 1

Reviewer 1 Report

Comments and Suggestions for Authors

This article reports a very surprising result, which should be of interest to all cancer researchers. The figures are interesting and informative. The number of attempts is sufficient. My questions are the following:

1. What exactly were the 7 normal and 6 carcinoma cells?

2. Was cell adhesion investigated?

Author Response

We thank the reviewer very much for taking the time to review this manuscript. Please find the detailed responses below and the corresponding revisions/corrections in track changes of the revised uploaded files.

Reviewer's Comment 1:

1. What exactly were the 7 normal and 6 carcinoma cells?

Response to Reviewer's Comment 1: We appreciate the reviewer giving us the opportunity to further clarify our results. We selected 7 human bronchial epithelial cell lines primary (UI) and immortalized cells with CDK4 and hTERT (KT) (HBEC3KT, HBEC14UI, HBEC14KT, HBEC30UI, HBEC30KT, HBEC34KT, HBEC56KT). The  6 human lung epithelial cancer cell lines (A549, NCIH460, NCIH520, HCC2352, HCC2814, HCC4017). We have updated Figure 1 and included a more detailed description of these cell lines via the following description inserted at Line 160 : We screened nonmalignantprimary and immortalizedlung bronchial epithelial cell lines previously established from biopsies of patients with and without lung cancer [Ramirez RD, Can Res. 2004; Sato M, Mol Can Res. 2013]. Primary cells were cultured and transfected with cyclin-dependent kinase 4 (CDK4) and human telomerase reverse transcriptase (hTERT), resulting in cultures that did not undergo senescence. Immortalized cells did not undergo malignant transformation in vitro due to their inability to form colonies in soft agar, nor in vivo did they form tumors in immune-compromised mice [Ramirez RD, Can Res. 2004]. 

Reviewer's Comment 2:

2. Was cell adhesion investigated?

Response to Reviewer's Comment 2: We thank the reviewer for this comment.  We understand the importance of extracellular matrix (ECM), and adhesion of the cell to the ECM is a key step in their migration through tissues. During the initial optimization of screening conditions, we only tested, the ability of cells to migrate on plastic, glass slides, collagen, and Matrigel. We plan to integrate cell adhesion into our continued investigation of cell migration/invasion in further studies.

Reviewer 2 Report

Comments and Suggestions for Authors

Dear Author,

I have carefully reviewed your article titled "Unexpected differences in the speed of non-malignant versus malignant cell migration reveal differential basal ATP levels". Overall, the study provides valuable insights into the differences in migration speed between non-malignant and malignant cells, attributing the disparities to variances in their basic ATP levels. However, I have identified several areas where the manuscript could be improved to enhance clarity and scientific rigor. In this review, I will provide detailed feedback on each point, along with suggestions for revision.

1. Differentiating non-malignant cells from malignant cells:

The article lacks a clear definition and criteria for distinguishing between non-malignant and malignant cells. It is important to establish the basis for determining malignancy, whether it is based on tumorigenicity or other biological behaviors such as proliferation, migration, or invasion. I recommend clarifying this aspect by explicitly stating the criteria used to differentiate non-malignant and malignant cells. This clarification will help readers understand the context and significance of the study.

2. Evaluating cell migration using multiple methods:

The main approach used in the study to quantify cell migration is the in vitro scratch assay. However, this method may produce false-positive results due to cell proliferation. Since cell migration is a key aspect of your study, relying solely on one method lacks persuasiveness. I suggest either incorporating the Transwell assay as an additional method to assess cell migration or refining the scratch assay to eliminate the influence of cell proliferation. For example, you could consider using serum-free or low-serum media (<2%) or pretreating the cells with mitomycin C (1µg/ml) to inhibit cell division. This will strengthen the validity of your findings and provide a more comprehensive analysis of cell migration.

3. Completeness of experimental data:

In the Results section, HBEC30UI (30UI) cell line is mentioned in sections 2, 3, and 5, but its migration ability is not assessed in the initial part of the results. This omission seems abrupt, and I recommend conducting preliminary experiments on this cell line to provide a more comprehensive and rigorous study. Adding the migration ability of HBEC30UI (30UI) cells to your results will enhance the completeness of your data and strengthen the overall findings.

4. Detailed explanation of media selection:

The study demonstrates that normal cells outperform cancer cells regardless of the medium used. However, the rationale behind selecting different types or concentrations of media, such as KSFM, KSFM 2%, and RPMI 5%, is not adequately explained. It remains unclear whether other types or concentrations of media would yield similar results. I suggest providing a more detailed explanation for the choice of these media or supplementing the study with additional experimental data using different types or concentrations of media. This additional information will provide readers with a better understanding of the experimental design and the impact of different media on cell migration.

5. Exploring other factors influencing cell migration:

The study suggests that physical parameters alone cannot account for the differences in migration speed, indicating the involvement of chemical factors. The hypothesis regarding ATP as a key chemical factor is supported by the results presented in sections 4 and 5. However, it is worth investigating whether other chemical factors also contribute to these differences in cell migration. I recommend citing relevant literature or including additional experimental data to elucidate the role of other chemical factors in cell migration. Additionally, the hypothesis regarding the role of the potential difference is not further elaborated upon. I suggest providing additional information and engaging in a more in-depth discussion in the corresponding section. Exploring other chemical factors and discussing the potential difference in more detail will enrich your study and provide a broader perspective on the factors influencing cell migration.

In conclusion, your article provides valuable insights into the differences in migration speed between non-malignant and malignant cells. By addressing the points outlined above, clarifying definitions, incorporating additional migration assays, ensuring completeness of experimental data, providing detailed explanations for media selection, and exploring the involvement of other chemical factors, your manuscript will be strengthened both in terms of scientific rigor and clarity. I hope you find this feedback helpful, and I look forward to seeing a revised version of your manuscript.

Comments on the Quality of English Language

The writing of this article is fluent and the use of words is appropriate, but please adjust individual words according to the content of the expression

Author Response

We thank the reviewer very much for taking the time to review this manuscript. Please find the detailed responses below and the corresponding revisions/corrections in track changes of the revised uploaded files.

Reviewer's Comment 1:

  1. Differentiating non-malignant cells from malignant cells:

The article lacks a clear definition and criteria for distinguishing between non-malignant and malignant cells. It is important to establish the basis for determining malignancy, whether it is based on tumorigenicity or other biological behaviors such as proliferation, migration, or invasion. I recommend clarifying this aspect by explicitly stating the criteria used to differentiate non-malignant and malignant cells. This clarification will help readers understand the context and significance of the study.

Response to comment 1:

We thank the reviewer for their comment and for giving us the opportunity to clarify our manuscript. We have cited and now incorporated into the manuscript the "clear definition and criteria for distinguishing between non-malignant and malignant cells" and " establish the basis for determining malignancy, whether it is based on tumorigenicity" with the following description inserted at Line 160 : We screened nonmalignantprimary and immortalizedlung bronchial epithelial cell lines previously established from biopsies of patients with and without lung cancer [Ramirez RD, Can Res. 2004; Sato M, Mol Can Res. 2013 Gao B, Oncotarget 2017]. Primary cells were cultured and transfected with cyclin-dependent kinase 4 (CDK4) and human telomerase reverse transcriptase (hTERT), resulting in cultures that did not undergo senescence. Immortalized cells did not undergo malignant transformation in vitro due to their inability to form colonies in soft agar, nor in vivo did they form tumors in immune-compromised mice [Ramirez RD, Can Res. 2004]. 

Reviewer's Comment 2:

  1. Evaluating cell migration using multiple methods:

The main approach used in the study to quantify cell migration is the in vitro scratch assay. However, this method may produce false-positive results due to cell proliferation. Since cell migration is a key aspect of your study, relying solely on one method lacks persuasiveness. I suggest either incorporating the Transwell assay as an additional method to assess cell migration or refining the scratch assay to eliminate the influence of cell proliferation. For example, you could consider using serum-free or low-serum media (<2%) or pretreating the cells with mitomycin C (1µg/ml) to inhibit cell division. This will strengthen the validity of your findings and provide a more comprehensive analysis of cell migration.

Response to comment 2:

We appreciate this comment as this is a question we receive often and should have been made clearer in our manuscript. All the experiments in the manuscript were developed for high-throughput assays on the Incucyte SX5 live-cell imager and analysis system which allowed us to follow cellular activities in real-time including but not limited to cell migration/ invasion, proliferation, and live cell metabolomics or the Livecyte kinetic cytometer system. We have now included some of the pilot experiments done to optimize the duration of our migration screening experiments in Figure 1d.  During our optimization of cell densities for ideal migration specifically times that did not intersect with proliferation, we noticed the initial observations reported in the manuscript. We have updated Figure 1 to include these data in figures, legends, and text- "These isogeneic cells were chosen not only for their genetic similarity but also for their close doubling times. The timing of migration experiments was designed to exclude gross complications due to proliferation (Figure 1d)."

Reveiwer's Comment 3:

  1. Completeness of experimental data:

In the Results section, HBEC30UI (30UI) cell line is mentioned in sections 2, 3, and 5, but its migration ability is not assessed in the initial part of the results. This omission seems abrupt, and I recommend conducting preliminary experiments on this cell line to provide a more comprehensive and rigorous study. Adding the migration ability of HBEC30UI (30UI) cells to your results will enhance the completeness of your data and strengthen the overall findings.

Response to comment 3:

We thank the reviewer for their comment and for giving us the opportunity to provide a more comprehensive and rigorous manuscript. We have included the answer to "Adding the migration ability of HBEC30UI (30UI) cells to your results" to Figure 1, and the text via the following: "Experimentally, we chose to focus on the isogeneic lung cell lines isolated from the same person. The HCC4017, cells were derived from the tumor burdened lung, while the primary HBEC30UI (30UI) and immortalized HBEC30KT (30KT), from the normal contralateral lung of the same patient23. These isogeneic cells were chosen not only for their genetic similarity but also for their close doubling times. The timing of migration experiments was designed to exclude gross complications due to proliferation (Figure 1d). Interestingly we found that both non-malignant cell lines, 30KT and 30UI, migrated significantly faster than the cancer cell line HCC4017 of the same person(Figure 1e and 1f). "

Reviewer's Comment 4:

  1. Detailed explanation of media selection:

The study demonstrates that normal cells outperform cancer cells regardless of the medium used. However, the rationale behind selecting different types or concentrations of media, such as KSFM, KSFM 2%, and RPMI 5%, is not adequately explained. It remains unclear whether other types or concentrations of media would yield similar results. I suggest providing a more detailed explanation for the choice of these media or supplementing the study with additional experimental data using different types or concentrations of media. This additional information will provide readers with a better understanding of the experimental design and the impact of different media on cell migration.

Response to comment 4:

We thank the reviewer for the opportunity for further clarification. We have included a more detailed explanation of our rationale for the use of the KSFM and RPMI 1640 with the following description inserted at Line 215: "RPMI 1640 growth media (supplemented with 5% fetal bovine serum, FBS) was used for maintenance of all our lung cancer cell lines, while all our normal bronchial epithelial cells were maintained in keratinocyte-serum free media (KSFM).  Although our initial screen for migration potential was conducted in cancer cells grown in RPMI 1640 supplemented with 5%FBS and normal cells in KSFM, we acknowledge that different media have vastly different components. Thus, in order to normalize extrinsic factor contribution to the varying cell speeds we adapted our lung cancer cells to KSFM with varying concentrations of FBS. While KSFM with no FBS supported HCC4017 migration cells did not proliferate, HCC4017 cells grown in KSFM plus 2% FBS supported proliferation and migration (Figure 3a). Interestingly, we found normal cells adapted for migration experiments in KSFM plus 2% FBS, or RPMI 1640 5% FBS were still able to close wounds (albeit, not statistically significant in RPMI 1640 with 5% FBS) faster (Figure 3b). Closer examination of wound closure dynamics indicated variations of both normal and cancer cells in RPMI 1640 5% FBS (Figure 3b-c). "

Reviewer's Comment 5:

  1. Exploring other factors influencing cell migration:

The study suggests that physical parameters alone cannot account for the differences in migration speed, indicating the involvement of chemical factors. The hypothesis regarding ATP as a key chemical factor is supported by the results presented in sections 4 and 5. However, it is worth investigating whether other chemical factors also contribute to these differences in cell migration. I recommend citing relevant literature or including additional experimental data to elucidate the role of other chemical factors in cell migration. Additionally, the hypothesis regarding the role of the potential difference is not further elaborated upon. I suggest providing additional information and engaging in a more in-depth discussion in the corresponding section. Exploring other chemical factors and discussing the potential differences in more detail will enrich your study and provide a broader perspective on the factors influencing cell migration.

Response to comment 5:

We appreciate this comment as this is a question that has fueled novel and further extensive study in our laboratory. We also appreciate the broad terminology and are currently investigating several other intrinsic and extrinsic genetic, chemical, and metabolic factors that may also account for the differences in migration speed we observe. We also greatly appreciate the chance to further clarify our current study and distinguish it from past studies. Here we focus on the fluctuations of intracellular ATP levels as a result of intrinsic differences. We have included " a more in-depth discussion in the corresponding section" to highlight the difference between previous studies and our own in the title and discussion to include terminology regarding intrinsic intracellular ATP comparisons of non-scratched to wound-induction with the following description inserted at Line 361: "Here, we report that normal lung epithelial cells have higher intrinsic intracellular basal and wound-induced intracellular ATP levels than lung tumor cells. There have been several studies examining the consequences of ATP release and generation following matrix detachment, wounding/injury or metastasis5,6,17,37 of normal or cancer cell. Our study, however, focuses on the high-throughput comparison of pre- and post-wound induction in normal cells side-by-side with cancer cell counterparts with the hope of identifying intrinsic differences in migratory potential."   

Reviewer 3 Report

Comments and Suggestions for Authors

The authors used primary cultures of lung and breast cells as examples to compare the motor potential of normal tissue cells and tumor tissue cells in their paper. Tumor cells are assumed to spread due to their superior migratory ability when compared to normal organ tissue. The search for ways to influence not only the proliferative potential but also the migratory (metastatic) activity is a significant challenge in both basic research and the development of novel chemicals capable of regulating this component of tumor growth.

The methodological approach to assessing the migration activity of various types of primary cultures of lung and breast cells is based on well-established methods of assessing horizontal (wound healing test) and vertical (migration of cells into the thickness of components mimicking extracellular matrix) migration, which are used to assess the functional potential of tumor cells in the presence of cytostatics. The evaluation of the energy potential of initial cell cultures is another intriguing technique. Both the researchers and the reviewer of this publication were surprised by the results produced by the writers. It was discovered that the primary culture of lung cells without tumor transformation had better mobility and energy potential than the primary culture of lung tumor cells.

Primary cultures of healthy lung tissue have a distinct starting level of energy potential and more motility, which is truly unusual in tumor biology.

The data acquired allowed the authors to reach a right conclusion about the outcomes of a comparative investigation of cell mobility and likely causes.

The citation list can be enhanced by removing publications older than 7 years.

The work's drawbacks include: 1) the poor choice of presenting the center and measure of tendency in the form of mean and standard mean error, when it is generally accepted to show this in the form of mean and standard deviation, which provides a more accurate picture of the differences between groups; and 2) the abundance of citation of the cited literature without an upper apostrophe (lines 235, 236, 275, 276).

The following questions were posed to the writers regarding the article:

1) What is the rationale for employing alternative nutrient media (RPMI-1640 and keratinocyte serum-free media kit) - to rule out the effect of the RPMI-1640 FBS medium supplement, which has an abundance of physiologically active substances - on the functional activity of the RPMI-1640 FBS medium supplement? So, what is the point of using a smaller volume of serum (2%) in a lot of experiments? 2) After discovering differences in the migratory activity of primary cultures of lung tissue, the scientists set out to discover differences in the energy potential of cells. However, they only looked at ATP, leaving the role of oxidative phosphorylation, glycolysis, and glutaminolysis as other sources of energy supply to lung tissue unanswered.  3) What role does fatty acid metabolism play in cell energy balance?

Author Response

We thank the reviewer very much for taking the time to review this manuscript. Please find the detailed responses below and the corresponding revisions/corrections in track changes of the revised uploaded files.

Reviewer's Comment 1: The work's drawbacks include: 1) the poor choice of presenting the center and measure of tendency in the form of mean and standard mean error, when it is generally accepted to show this in the form of mean and standard deviation, which provides a more accurate picture of the differences between groups; and 2) the abundance of citation of the cited literature without an upper apostrophe (lines 235, 236, 275, 276).

1) We thank the reviewer for aiding us in providing a more accurate statistical analysis. We have re-analyzed all our data regarding speed differences of lung cells in Figures 1, Figure 4, and Figure 5 to reflect standard deviation. 

2)We thank the reviewer for bringing this formatting error to our attention. We have fixed and reformatted our bibliography to include the proper formatting and citations.

Reviewer's Comment 2: 1) What is the rationale for employing alternative nutrient media (RPMI-1640 and keratinocyte serum-free media kit) - to rule out the effect of the RPMI-1640 FBS medium supplement, which has an abundance of physiologically active substances - on the functional activity of the RPMI-1640 FBS medium supplement? So, what is the point of using a smaller volume of serum (2%) in a lot of experiments?

Response to Reviewer Comments 2: 

We thank the reviewer for the following suggestions and have included the following description inserted at Line 215: "RPMI 1640 growth media (supplemented with 5% fetal bovine serum, FBS) was used for maintenance of all our lung cancer cell lines, while all our normal bronchial epithelial cells were maintained in keratinocyte-serum free media (KSFM).  Although our initial screen for migration potential was conducted in cancer cells grown in RPMI 1640 supplemented with 5%FBS and normal cells in KSFM, we acknowledge that different media have vastly different components. Thus, in order to normalize extrinsic factor contribution to the varying cell speeds we adapted our lung cancer cells to KSFM with varying concentrations of FBS. While KSFM with no FBS supported HCC4017 migration cells did not proliferate, HCC4017 cells grown in KSFM plus 2% FBS supported proliferation and migration (Figure 3a). Interestingly, we found normal cells adapted for migration experiments in KSFM plus 2% FBS, or RPMI 1640 5% FBS were still able to close wounds (albeit, not statistically significant in RPMI 1640 with 5% FBS) faster (Figure 3b). A closer examination of wound closure dynamics indicated variations of both normal and cancer cells in RPMI 1640 5% FBS (Figure 3b-c). "

Reviewer's Comment 3: After discovering differences in the migratory activity of primary cultures of lung tissue, the scientists set out to discover differences in the energy potential of cells. However, they only looked at ATP, leaving the role of oxidative phosphorylation, glycolysis, and glutaminolysis as other sources of energy supply to lung tissue unanswered.  3) What role does fatty acid metabolism play in cell energy balance?

Response to Reviewer Comments 3: 

We appreciate this comment as this is a question that has fueled novel and further extensive study in our laboratory.  We also greatly appreciate the chance to further clarify our current study and distinguish it from past studies. Here we focus on the fluctuations of intracellular ATP levels as a result of intrinsic differences. We have begun investigating several other extrinsic genetic, chemical, and metabolic factors that may also account for the differences in migration speed we observe in this study. We appreciate the reviewer's specific suggestions ", they only looked at ATP, leaving the role of oxidative phosphorylation, glycolysis, and glutaminolysis as other sources of energy supply to lung tissue unanswered.  3) What role does fatty acid metabolism play in cell energy balance?" our studies of glucose and fat oxidation have yielded interesting metabolic consequences for the cells during migration. As mentioned previously, these metabolic experiments form the basis for a larger study of extrinsic factor regulation of migration. Specifically, we have initiated experimentation of several pathways that are parallel and key to our current study but unfortunately outside the constraints of our current manuscript. We have also chosen to highlight the difference between previous studies and our own in the title and discussion to include terminology regarding intrinsic intracellular ATP comparisons of non-scratched to wound-induction with the following description inserted at Line 361: "Here, we report that normal lung epithelial cells have higher intrinsic intracellular basal and wound-induced intracellular ATP levels than lung tumor cells. There have been several studies examining the consequences of ATP release and generation following matrix detachment, wounding/injury or metastasis5,6,17,37 of normal or cancer cell. Our study, however, focuses on the high-throughput comparison of pre- and post-wound induction in normal cells side-by-side with cancer cell counterparts with the hope of identifying intrinsic differences in migratory potential."